# Killing Two Birds with One Stone: Upgrading Organic Compounds via Electrooxidation in Electricity-Input Mode and Electricity-Output Mode

**DOI:** 10.3390/ma16062500

**Published:** 2023-03-21

**Authors:** Jiamin Ma, Keyu Chen, Jigang Wang, Lin Huang, Chenyang Dang, Li Gu, Xuebo Cao

**Affiliations:** 1College of Biological, Chemical Sciences and Engineering, Jiaxing University, Jiaxing 314001, China; 2School of Chemistry and Chemical Engineering, Shandong University of Technology, Zibo 255049, China; 3School of Materials and Textile Engineering, Jiaxing University, Jiaxing 314001, China

**Keywords:** electrochemically oxidative upgrading reaction (OUR), hydrogen production, carbon dioxide electroreduction, fuel cell

## Abstract

The electrochemically oxidative upgrading reaction (OUR) of organic compounds has gained enormous interest over the past few years, owing to the advantages of fast reaction kinetics, high conversion efficiency and selectivity, etc., and it exhibits great potential in becoming a key element in coupling with electricity, synthesis, energy storage and transformation. On the one hand, the kinetically more favored OUR for value-added chemical generation can potentially substitute an oxygen evolution reaction (OER) and integrate with an efficient hydrogen evolution reaction (HER) or CO_2_ electroreduction reaction (CO_2_RR) in an electricity-input mode. On the other hand, an OUR-based cell or battery (e.g., fuel cell or Zinc–air battery) enables the cogeneration of value-added chemicals and electricity in the electricity-output mode. For both situations, multiple benefits are to be obtained. Although the OUR of organic compounds is an old and rich discipline currently enjoying a revival, unfortunately, this fascinating strategy and its integration with the HER or CO_2_RR, and/or with electricity generation, are still in the laboratory stage. In this minireview, we summarize and highlight the latest progress and milestones of the OUR for the high-value-added chemical production and cogeneration of hydrogen, CO_2_ conversion in an electrolyzer and/or electricity in a primary cell. We also emphasize catalyst design, mechanism identification and system configuration. Moreover, perspectives on OUR coupling with the HER or CO_2_RR in an electrolyzer in the electricity-input mode, and/or the cogeneration of electricity in a primary cell in the electricity-output mode, are offered for the future development of this fascinating technology.

## 1. Introduction

The overuse of fossil fuels makes the economy susceptible to supply bottlenecks and price spikes and aggravates the greenhouse effect and environmental pollution worldwide. Recently, the rapid development of renewable-based electricity demonstrates great potential in mitigating the energy crisis. Unfortunately, renewable-based electricity such as wind power and solar power is only produced when the wind is blowing or the sun is shining; thus, it cannot effectively connect to the power grid. The introduction of green electricity, based on renewables, troubles direct utilization via the power grid. Therefore, it is imperative to develop clean and renewable alternatives for energy storage and chemical conversion. With the remarkable renaissance of electrochemical technology, and the constant inventions in the rapid development of renewable-based electricity technology, the technology of energy storage and conversion into chemicals is attracting great interest [1,2,3,4,5,6]. For example, hydrogen, a clean, flexible, cost-effective energy carrier for net-zero carbon (carbon-free) strategies, features a higher enthalpy of combustion (H_2_(g) + ½O_2_(g) → H_2_O(l) ΔH = −285.8 kJ/mol, high specific-energy-density) and enables the possibility of attaining secure and clean energy in the future. From the perspective of hydrogen production, conventional hydrogen production technology based on fossil fuels is energy-intensive and environmentally unfriendly. Meanwhile, water electrolysis for hydrogen generation has emerged and is promising because of mild reaction conditions and zero-carbon emissions (coupled with green electricity). In particular, with society progressing into renewable-based economics, water electrolysis enables the storing of electricity from intermittent and renewable wind power, solar power and nuclear power into chemicals. This wise strategy opens up great opportunities for the development of enormous energy storage capacities for excess electricity originating from renewable energy sources with the rapid increase in global energy consumption. Over the course of water electrolysis, hydrogen is generated (HER) in a cathodic chamber and oxygen is evolved (OER) in an anodic chamber. Unfortunately, the OER suffers from a relatively high theoretical oxidation potential and sluggish kinetics that severely affect the nominal efficiency of water electrolysis. Moreover, the oxygen diffusing into the cathode chamber can be reduced back to water, which reduces the overall efficiency of the process. In alkaline water electrolysis, the extensive hydrogen crossover permeation into the opposite chamber generates safety concerns (low explosive limit of 4% mol H_2_) [7]. In a proton exchange membrane (PEM) cell, the coexistence of the H_2_, O_2_ and Pt catalysts was observed to generate hazardous reactive oxygen species (ROS), such as hydroxyl (HO•) and hydroperoxyl radicals (HOO•), which can result in an oxidative membrane backbone degradation and side chain disintegration, thereby shortening the lifetime of PEM [8].

Recently, the kinetically favored OUR of organic compounds has been demonstrated to effectively substitute the OER to integrate with the HER in a hybrid electrolyzer (OUR||HER), which can not only generate value-added chemicals anodically for the potential manufacturing of pharmaceuticals, fine chemicals, food, pesticides, materials, etc., but also produce hydrogen cathodically [2,9,10,11,12,13,14,15,16]. In comparison to conventional water electrolysis, the OUR||HER mode requires reduced cell voltages, is more energy-saving and cost-effective and has a theoretically high utilization of electricity. Furthermore, the OUR||HER mode is more advanced than the sacrificial agent (e.g., urea [17,18], hydrazine [19,20], ammonia [21], et al.)-assisted hydrogen production that requires the constant consumption of the sacrificial agent, thereby increasing the capitalized cost, especially for large-scale applications. Notably, from the perspectives of wastes removal, there is no need to take the price of the sacrificial agent (wastes) into account over the process of environmental remediation. Instead, the additional benefit of harmful pollutants’ decomposing can be achieved as well as the hydrogen production [22].

Besides OUR||HER, OURs of organic compounds have been reported to couple with the carbon dioxide electroreduction reaction (CO_2_RR). The CO_2_RR can not only potentially curb CO_2_ emissions, but also generate value-added chemicals, which, however, are still limited to the extremely stable chemical bond in CO_2_ (C=O, 806 kJ mol^−1^), complicated reaction pathway, resulting poor activity and selectivity and sluggish anodic OER process. Therefore, the OUR of organic compounds is an appealing strategy to substitute the OER for effective CO_2_RRs (OUR||CO_2_RR) (Figure 1, left). It seems that the coupling strategy of OUR||HER, OUR||CO_2_RR and/or other analogous pairing electrolysis is a fascinating technology that can kill two birds with one stone. Up to now, alcohols, aldehydes, amines, nitroalkanes, organic sulfides, alkenes, etc., have been reported for efficient OUR||HER and/or OUR||CO_2_RR [23,24,25,26,27,28,29,30,31], and could even probably couple with other important reduction reactions such as nitrogen fixation (NRR) and nitrate electro-reduction. This strategy is such an ideal electrochemical process that target products with high added value are generated at both anodes and cathodes, thus possessing great potential to enhance the utilization rate of electricity and lead to economic benefits. This kind of paired electrolysis exhibits positive aspects such as much more reduced cell voltage (reducing the energy consumption), high theoretical energy efficiency and atom economy, no production of ROS or explosive hazards, etc.

Meanwhile, the OUR can be realized via the electrooxidation reactions in a primary cell or battery from the electricity-input mode to the electricity-output, such as fuel cells and zinc–air batteries fed by biomass-derived carbonaceous fuel, as displayed in Figure 1 (right) [11,32,33]. The oxidation reaction is also the basic unit of fuel cells that possess the merits of high efficiency, environmental friendliness and no external charging. Furthermore, traditional carbonaceous-based fuel cell technology generates low-valued products of CO_2_ and or CO_3_^2−^ that are not conducive to the realization of the goal of reducing peak emissions of carbon dioxide and carbon neutrality. In addition to alleviating the issue of energy shortages, the OUR of organic compounds in fuel cells facilitates the green and profitable cycle of waste biomass-derived resources and cuts the carbon footprint [32]. For example, glycerol manufactured from animal fats and vegetable oils can be utilized as fuel in a fuel cell with value-added product generation, which is more environmentally friendly than conventional fuel cells [32]. The OUR of organic compounds in fuel cells seems to be another fascinating strategy for electrosynthesis for both chambers (with oxygen being reduced to hydroxy peroxide) and electricity. Up to now, various alcohols (e.g., ethanol, ethylene glycol, isopropanol, glycerol and Furfural) [32,34,35,36,37] have been demonstrated to have great potential for effectively upgrading fuel cells. Besides fuel cells, upgrading organic compounds can also be realized in other devices, such as zinc–air batteries [33]. From the aforementioned description, organic compounds enabling value-added chemical generation can be realized via an electrolyzer, namely the electricity-input mode, as well as a primary cell (such as a fuel cell), namely the electricity-output mode.

This minireview intends to highlight the recent developments and landmarks of the OUR of organic compounds for value-added product generation to integrate with the HER or CO_2_RR in a hybrid electrolyzer and in batteries or cells for electricity generation, in terms of catalysts, performance and the structure-performance relationship. Additionally, challenges and perspectives for the future development of oxidative upgrading of organic compounds in parallel with hydrogen production or the CO_2_RR in a hybrid electrolyzer in electricity-input mode, and/or in a fuel cell in electricity-output mode are discussed. We hope the perspectives we have outlined in this minireview will proceed in order to develop efficient electrolyzers via the electricity-input mode and/or fuel cells (or other analogous devices) of the electricity-output mode as a cost-effective and robust solution to help in realizing the targets of effective electrosynthesis and carbon neutrality.

## 2. Anodic OUR of Organic Compounds in Parallel with HER (OUR||HER)

### 2.1. Anodic OUR of Alcohol Integrating with Cathodic HER

The OUR of methanol integrates with the HER. Compared to the traditional OER, the anodic OUR of organic molecules reduces the working voltage and possesses faster reaction kinetics [3,26]. As the simplest alcohol, methanol enables easy production via chemical or biomass industrial synthesis. Industrial methanol is inexpensive (about EUR 350 per tonne); meanwhile, methanol exhibits remarkable solubility in water. Furthermore, the methanol electrooxidation reaction (MOR) produces value-added formate salts or formic acid (near EUR 539 per tonne) and possesses fast kinetics. Thus, the OUR of methanol to replace the OER is a promising alternative for hydrogen production and chemical conversion (Table 1) [38,39,40,41,42,43]. Noble metal catalysts such as Pt and Pd have been demonstrated to be effective for the active electrooxidation of alcohols; however, these noble metal catalysts are limited by their low quantities, high price and easy CO-poisoning in alkaline medium that impedes reaction kinetics [32,44,45]. Precious metal electrocatalysts such as Ni, Co-based nanostructures have been demonstrated to be potential candidates for efficient alcohol electrocatalysts owing to their weak adsorption of CO and low price [33,43]. Liu and co-workers [43] constructed a three-dimensional Mo-doped Ni(OH)_2_ with increased density of active sites and quite low Ni–Ni coordination (Figure 1a), which can jointly boost the MOR activity yielding formate (Figure 1b). Meanwhile, the hybrid electrolyzer of methanol upgraded for formate generation by Mo-Ni(OH)_2_ anodically, and its integration with hydrogen production by Ni_4_Mo–MoO_2_ cathodically, display a relatively lower cell voltage of 1.52 V at 130 mA cm^−2^ and good stability (Figure 1c,d). Furthermore, an industrial level electrolysis was boosted with a decent current density of more than 500 mA cm^−2^ under industrial conditions (cell voltage of 2.00 V, 6 M KOH), and high selectivity of above 90%, signifying great potential for industrial application (Figure 1e–g). DFT results suggested that the excellent performance is mainly attributed to the ultralow Ni–Ni coordination effect for active Ni sites in Mo-Ni(OH)_2._ Notably, although methanol exhibits significant merits for upgrading with hydrogen production, designs of advanced catalysts and systems are still desirable, and methanol is toxic. Thus, attention should be paid to operation.

The OUR of ethanol integrates with the HER. Typically, aliphatic alcohol with low length of carbon chain exhibits relatively significant water solubility and relatively lower theoretical potential of oxidation, and thus shows great potential for oxidative upgrade coupling with hydrogen production. Ethanol derived from biomass fermentation has been demonstrated to be a promising alternative to renewable energy carriers and green chemicals. It serves as liquid fuel and a versatile feedstock for synthesizing fine chemicals such as acetic acid, ethyl acetate, ethylene, acetaldehyde and 1,1-diethoxyethane (DEE). For example, DEE serves as an important feedstock for the synthesis of pharmaceuticals, perfumes, polyacetal resins, etc. [27]. Unfortunately, the variety of product distribution poses great obstacles for the selective ethanol electrooxidation reaction (EOR) from the viewpoint of electrosynthesis. To achieve this goal, catalysts of fine design and synthesis, and a thorough understanding of mechanisms, are crucial. Although noble metal catalysts are limited to their high price and scarcity, they cannot be replaced for various reactions. Typically, pure Pt suffers from a weak CO anti-poisoning ability over alcohol electrooxidation. To solve this problem, an alloying strategy is often utilized to modulate the electronic structure of active sites, thus enhancing the CO anti-poisoning ability, and thereby boosting the alcohol electrooxidation activity. Alberto Rodríguez-Gómez et al. [46] developed a Pt-based bimetallic catalyst system (PtM) to boost hydrogen and value-added compound generation (acetic acid, acetaldehyde and ethyl acetate). They found that this secondary metal can influence the electrocatalytic performance as well as product distribution. For example, PtCo/C and PtNi/C possessed the highest electrochemical activity at high polarization levels. In comparison, a lower potential interval (<0.85 V) was needed for PtRu/C to promote the acetic acid production, despite sacrificing ethanol conversion. One-dimensional nanostructures such as nanowires and/or nanotubes seem to have great potential for efficient alcohol electrooxidation [32,44]. Guo and co-workers [26] reported a first example showcasing the highly efficient electro-generation of DEE in the anodic chamber (Faraday efficiency of DEE, 85%) coupling with high-purity hydrogen (highest-ever-reported Faraday efficiency of DEE, 94%) at the cathode using PtIr nanowires (denoted as PtIr NWs, diameter: about 1 nm) as the bifunctional catalysts, as shown in Figure 2a. They demonstrated that the hybrid electrolyzer of PtIr NWs||PtIr NWs achieved a lower voltage of 0.61 V at 10 mA∙cm^−2^ than those of the Pt NWs||Pt NWs electrolyzer (0.85 V) and the state-of-the-art commercial Pt/C||Pt/C electrolyzer (0.86 V) (Figure 2b–d). In situ infrared spectroscopy results implied that PtIr NWs is conducive to the effective activation of C−H and O−H bonds of ethanol molecules that are crucial for the formation of acetaldehyde intermediates and the final DEE (Figure 2e,f). Additionally, the hybrid electrolyzer with PtIr NWs as bifunctional catalysts delivered excellent stability without an obvious decrease of the Faraday efficiency over DEE generation. Although this kind of PtIr NWs catalyst exhibited the highest-ever-reported efficiency in pair electrolysis for ethanol electrooxidation and hydrogen production, the fact that catalysts were configurated with PGM-based catalysts cannot be neglected, which therefore presents the opportunity to develop robust and efficient noble-metal free catalysts in the future. Besides the PGM-based catalysts, earth-abundant metal catalysts have also been reported for an efficient EOR. Recently, Wang and co-workers [33] developed a heterostructured Co(OH)_2_@Ni(OH)_2_ catalyst for efficient and selective ethanol electrooxidation to acetate (1.30 V vs. RHE at 10 mA cm^−2^, FE: 97.9%). A relatively lower potential of the symmetric Co(OH)_2_@Ni(OH)_2_||Pt/C electrolyzer was needed (1.39 V vs. RHE at 10 mA cm^−2^). Although the OUR of ethanol possesses the advantages of being non-toxic, promoting flexible product distributions and showing great potential to integrate with hydrogen production, the volatility of ethanol probably hinders its practical application to some extent, and the flexible product distribution poses high requirements for catalysts and purification.

The OUR of glycerol integrates with the HER. Glycerol, a biproduct of biodiesel manufactured from vegetable oils and animal fats, has increased with an average annual growth rate of about 4.1% worldwide over the last decade [32]. Therefore, efficient utilization of glycerol is urgent and profitable. Up to now, the main strategy of utilizing glycerol has consisted of catalytic oxidation to yield value-added chemicals, such as 1,3-dihydroxyacetone, glyceraldehyde, glyceric acid and formate, which are widely applied in the pharmaceutical, food, and cosmetics industries. Glycerol electrochemical oxidation (GOR) is used in smart devices such as electrolyzers and fuel cells, which are emerging as promising platforms for the clean utilization of this alcohol [32,47,48,49,50,51]. Recently, Huang and co-workers [32] developed bifunctional PdPtAg nanowires via facile galvanic displacement reaction. The as-prepared catalysts possessed low peak-potential of the GOR oxidation at near 0.9 V vs. RHE and a relatively high formate selectivity of 81.2%. These significant merits endow the PdPtAg nanowire-based catalysts with great potential for the application of GOR-promoted hydrogen production. Recently, Chen and co-workers [50] found that GOR can be boosted via modulating proton and oxygen anion deintercalation in NiCo hydroxide, as manifested by the DFT and experimental characterizations (Figure 3a–c), which displayed a relatively low potential of 1.35 V at 100 mA cm^−2^, and a formate selectivity of 94.3% of GOR in a half-cell (Figure 3d). Furthermore, 1.33 and 1.58 V are required in a hybrid electrolyzer (NiCo hydroxide||NiCo hydroxide) to reach 10 and 100 mA cm^−2^ in GOR-promoted hydrogen production (Figure 3e,f).

Other aliphatic alcohol upgrading can also be integrated with hydrogen production in “pared electrolysis”, such as electrooxidation of ethylene glycol, isopropanol and 1,3-propandiol [52,53,54,55,56,57]. Besides the aforementioned aliphatic alcohols, some alcohols with aromatic groups and relatively good solubility of water can also be effectively coupled with the hydrogen evolution reaction, such as benzyl alcohol and its oxidized product, benzoic acid, which is a basic fine chemical utilized in synthetic fiber, resin and in the antiseptic industries [58,59,60,61]. Duan and co-workers developed a composited catalyst of Au nanoparticles supported on Co oxyhydroxide nanosheets (Au/CoOOH) for benzyl alcohol upgrade integrating with H_2_ production at 1.5 V vs. RHE with a current density of 540 mA cm^−2^. The absolute current can approach 4.8 A at 2.0 V in a hybrid flow electrolyzer. The experimental analysis combined with the theoretical calculations indicated that the benzyl alcohol can be enriched at the Au/CoOOH interface and oxidized by the electrophilic oxygen species (OH*) generated on CoOOH, resulting in higher activity than pure Au.

### 2.2. Anodic OUR of Aldehyde Integrating with Cathodic HER

Besides alcohol electrooxidation, electrochemically oxidative upgrading of biomass and biomass-derived platform molecules, such as aldehydes, also enables the generation of value-added chemicals and integration with the hydrogen evolution reaction in a hybrid electrolyzer. 5-hydroxymethylfurfural (HMF) is a vital building block for the pharmaceutical and other chemical industries. Its oxidized derivative of 2, 5-furandicarboxylic acid (FDCA) has been reported to be a green platform chemical to substitute terephthalic acid as a feedstock for the preparation of high-value-added polymers [62]. HMF Electrooxidation is mostly performed under alkaline conditions wherein hydroxide ions are necessities. Therefore, electrocatalytic HMF oxidation enables an integration with the HER, producing an additional product with high economic value, thus increasing the energy efficiency of hydrogen production. The pioneering work of HMF electrooxidation to FDCA was presented by Grabowski and co-workers; the yield is 71% [63]. Platinum group metal-based catalysts have also been reported with relatively high activity and selectivity to replace transition-metal-based catalysts [24,64]. Cha and Choi [64] demonstrated a FE of 100% for FDCA at 1.54 V on a 2, 2, 6, 6-tetramethylpiperidine-1-oxyl (TEMPO)-mediated Au electrode. However, the TEMPO mediator increases the costs of downstream separation. Evidently, a high pH boosts the selective electrooxidation of HMF to FDCA [65]. At high pH values above 13, HMF is decomposed into humin-type products. Thus, to increase product yield, fine design of the catalyst system is of extreme importance, especially regarding the noble metal-free catalysts (e.g., earth-abundant metal oxides [23], nitrides [66], phosphides [67] and borides [68]). For instance, Stefan Barwe et. al. [68] developed a high-surface-area nickel boride (Ni_x_B) electrode with high FE of near 100% for the FDCA generation with a yield of 98.5%, and demonstrated HMF is preferentially oxidized via hydroxymethyl-2-furancarboxylic acid rather than 2, 5-diformylfuran through operando electrochemical–infrared spectroscopy that agrees with HPLC analysis. A relatively low potential of 1.45 V vs. RHE is achieved with a current density of 100 mA∙cm^−2^ over HMF electrooxidation, which is 170 mV lower than the potential necessary to evolve the OER at the same conditions. Other oxygen-contained molecules such as glucose, sodium gluconate and triclosan can also be electrooxidized to couple with the HER [69,70,71].

### 2.3. Anodic OUR of Carboxylates Integrating with Cathodic HER

Since the first pioneering report in the mid-nineteenth century, carboxylate electrooxidation or the Kolbe electrolysis has been extensively studied [72,73,74,75,76]. The classical manifestation of oxidative decarboxylation of an aliphatic carboxyl acid can generate a transient alkyl radical (RCO_2_•) which are then combined to produce a C*sp^3^*-C*sp^3^* bond [77]. Such a heterocoupling of C*sp^3^*-C*sp^3^* between two carboxylic acids was historically utilized as a key strategy to synthesize prostaglandin [78], jasomonic acid analogues [79], a modular route to access sugar derivatives [80] and vicinal olefin functionalizations [81], albeit to a lesser extent. Recently, Baran and co-workers developed a mildly reductive Ni-electrocatalytic strategy to integrate two different carboxylates through in situ generated redox-active esters, which was termed doubly decarboxylative cross-coupling. This operationally facile method enables heterocoupling of primary, secondary and even certain tertiary redox-active esters, thus generating a powerful new strategy for electrosynthesis. Furthermore, the reaction cannot be mimicked by applying either stoichiometric metal reductants or photochemical conditions, and it can tolerate a range of functional groups. Furthermore, this method was scalable for the synthesis of 32 compounds, which reduced overall step counts by 73%. Besides the desired Kolbe-product, a co-product of CO_2_ with a double-molar fraction was generated at the anode and usually stoichiometric amounts of H_2_ from water electrolysis at the cathode. Therefore, in an undivided electrolyzer, the respective gaseous electrolysis products of CO_2_ and H_2_ are to be obtained, and these can be directly utilized without further purification [82]. Recently, Markus Stöckl and co-workers [82] reported the first coupling of the anodic Kolbe electrolysis with a subsequent microbial conversion of the produced coproduct hydrogen at the cathode. Kolbe electrolysis of valeric acid yielded n-octane. Then, isopropanol was generated by resting Cupriavidus necator cells with gaseous products (CO_2_ and H_2_). The resting microbial cells exhibited Fes of 80% and 60% for cathodic and anodic reactions, respectively, and carbon efficiencies of up to 41%. The implementation of a hybrid electrolyzer resulted in striking process performance with overall efficiencies of up to 64.4%, as displayed in Figure 4.

### 2.4. Anodic OUR of Nitrogen-Contained Molecules Integrating with Cathodic HER

In addition to the aforementioned aldehydes and alcohols, amine, nitroalkane and tetrahydroisoquinoline electrooxidation are also kinetically favorable and can also generate value-added imines, amides, nitriles, azo compounds and amine oxides, which are widely utilized in the field of pharmaceuticals and agrochemicals [12,14,83,84]. In 2018, Zhang and co-workers [83] developed NiSe nanorod arrays for thermodynamically more favorable primary amine (−CH_2_−NH_2_) electrooxidation in water to replace the OER and couple the HER. The increased H_2_ generation can be realized at the cathode; meanwhile, the as-prepared NiSe nanorod arrays displayed significant diversity of various aromatic and aliphatic primary amines to selectively generate nitriles with decent yields (>93 %) and selectivity (>94 %) at the anode. Mechanistic investigations suggest that the reconstruction of nickel species (Ni^II^/Ni^III^ species) may serve as redox active sites for the transformation of primary amines. Meanwhile, the hydrophobic nitrile products enabled ready escape from the aqueous electrolyte/electrode interface, avoiding the deactivation of catalysts and thereby promoting continuous gram-scale synthesis. Later on, Zhai and co-workers [84] utilized the operando electrocatalysis variations (i.e., chalcogen leaching) to manipulate the electrocatalytic interface toward amine electrooxidation, as shown in Figure 5. Taking chalcogen-doped Ni(OH)_2_ as an example (Figure 5a–c), experimental results (e.g., XPS and in-situ Raman spectrum in Figure 5f,g) uncovered that chalcogens leach from the substrate and are then adsorbed on the NiOOH surface as chalcogenates over the electrooxidation process. The charged chalcogenates can induce a local electric field that drives the polar amines through the Helmholtz plane to enrich on the electrocatalytic surface. Meanwhile, the local polarization of chalcogenates and amines can boost dehydrogenated activation of amino C–N bond to nitrile C≡N bonds. Under the pushing effect of surface-adsorbed chalcogenate ions, the Ni(OH)_2_ displayed near-full propionitrile selectivity (99.5%) at a potential of 1.317 V vs. RHE with an ultrahigh current density, as displayed in Figure 5d,e,h.

### 2.5. Anodic OUR of Other Organic Molecules Integrating with Cathodic HER

Recently, two other kinds of organic molecule electrooxidation reactions have attracted chemists’ interest: electrooxidation of organic sulfides (SOR) and alkene (AEOR). (i) Regarding the electrooxidation of organic sulfides (SOR), organic sulfoxides and sulfone as the main products of sulfide oxidation play prominent roles in pharmaceuticals, biological processes and material science [85]. Unfortunately, the conventional oxidation of organic sulfides usually suffers from the required use of strong oxidizing agents (e.g., iodine, H_2_O_2_, 3-chloroperbenzoic acid, or peroxy acids and oxone), homogeneous catalysts, toxicity, etc. [3], whereas recent reports have demonstrated successful electrooxidation of sulfides using water as an oxygen source, and, e.g., MeCN, acetone and DMF as the solvent on carbon electrodes, NiII complexes, cobalt pentacyanonitrosylferrate and CoFe-LDH [86,87,88,89,90,91]. Zhang and co-workers [86] developed an electrocatalytic protocol to selectively oxidize sulfides to sulfoxides at cost-effective graphite felt electrodes. They used NaCl to serve as an electrolyte and a redox mediator to protect sensitive functional groups from oxidation. This metal-free electrocatalytic protocol is simple, green and compatible with different sensitive functional groups with acetone/water as the green solvent. The methodology could easily work on a gram or even a decagram scale, and it displays great potential regarding coupling with the hydrogen evolution reaction. (ii) Regarding the electrooxidation of alkene (AEOR), alkene oxidation is an effective strategy to synthesize vicinal diols and epoxides, which are key intermediates for the synthesis of perfumes, food additives, drug intermediates, fine chemicals and agrochemicals [92]. For example, Ethylene oxide is among the most abundantly fabricated commodity chemicals worldwide due to its wide application in the plastics industry, notably for fabricating polyesters and polyethylene terephthalates. In an early work pioneered by Meyer and co-workers in 1980 [93], cyclohexene electrooxidation was reported to produce p-benzoquinone with a Ru-based complex catalyst. Recently, Sargent group conducted a series of wonderful work on ethylene electrooxidation to produce ethylene glycol (80% selectivity) and ethylene oxide (EO), and this shows great promise to integrate with hydrogen evolution [94,95,96]. In 2020, they found that gold-doped palladium can oxidize ethylene to yield ethylene glycol with approximately 80% FE at ambient condition in aqueous media [94]. Later on in 2020 [95], they applied an extended heterogeneous:homogeneous interface (Figure 6a) and utilized chloride as a redox mediator at anode (Figure 6b) to boost the selective ethylene electrooxidation for ethylene oxide generation with a large current density of 1 A∙cm^−2^, high FE of ~70% and product specificities of ~97% (Figure 6c–f). In another work by the same group [96], they found that barium-oxide-loaded catalysts can obtain an ethylene-to-EO FE of 90%. This redox-mediated paired system exhibited a 1.5-fold higher CO_2_-to-EO FE (35%) and utilized a 1.2 V lower working voltage than the benchmark electrochemical systems.

**Table 1 materials-16-02500-t001:** Electrocatalytic performance of recently reported representative OUR electrocatalysts and the OUR||HER-based electricity-input system.

Working Mode	Reaction Type	Anode Catalyst	Substrate	Electrolyte	Product	3-Electrode System		2-Electrode System		Ref.
						E_OER_ at j_10_ (V_RHE_)	E_OUR_ at j_10_ (V_RHE_)	V_OER-HER_ at j_10_	V_OUR-HER_ at j_10_	
OUR||HER on electricity-input mode	Alcohol oxidation	Vp-Ni_2_P-Pt/CC	methanol	2 M in 1 M KOH	Formate/H_2_	0.72 at J_50_	1.651 at J_50_	-	ca. 0.7	[38]
		CeO_2_/RuO_2_	methanol	2.5 M in 0.5 M H_2_SO_4_	Formic acid/H_2_	1.495	1.195	1.568	1.308	[39]
		Co(OH)_2_@HOS	methanol	3 M in 1 M KOH	formate/H_2_	1.571	1.385	1.631	1.497	[41]
		PtIr NWs	ethanol	4 M in 0.5 M HClO_4_	DEE/H_2_	-	0.45	-	0.61	[26]
		Co(OH)_2_@Ni(OH)_2_	ethanol	1.0 M in 1.0 M KOH	acetate/H_2_	-	1.3	-	1.39	[33]
		Gold	glycerol	0.1 M in 0.1 M NaOH	glyceric acid/H_2_	-	1.0	-	-	[47]
		NiCo hydroxide	glycerol	0.1 M 1 M KOH	formate/H_2_	-	1.39 at J_100_	-	1.33	[50]
		PdAg/NF	ethylene glycol	1 M in 0.5 M KOH	glycolic acid/H_2_	1.55	0.57	-	1.02 at j_20_	[57]
		CoNC	glucose	0.1 M in 1.0 M KOH	gluconic acid, glucaric acid/H_2_	1.7 at J_100_	1.5 at J_100_	1.78 V at J_100_	0.9 V at J_100_	[69]
		Ni(OH)_2_	benzyl alcohol		benzoic acid/H_2_		~1.33 at J_100_			[58]
	aldehyde oxidation	NixB	HMF	10 mM in 1 M KOH	FDCA/H_2_	1.62 at J_100_	1.45 at J_100_	-	-	[68]
	carboxylate oxidation	NiCl_2_•dme, Ligand L4	carboxylic acids	NaI (0.2 M), DMF	decarboxylative products	-	-	-	4 mA, 4 F per mol	[77]
		Pt-foil	valeric acid	0.5 M	n-octane	-	-	-	-	[82]
	amine oxidation	NiSe	Benzyl-amine	1 mM in 1.0 M KOH	benzyl nitrile/H_2_	1.48	1.34	1.70 at J_20_	1.49 at J_20_	[83]
		t-Ni/Co MOF	Benzyl-amine	0.02 M in 1.0 M KOH	benzonitrile/H_2_	-	-	~1.75	~1.5	[14]
		S-Ni(OH)_2_	Propyl-amine	0.1 M in 1.0 M KOH	propionitrile/H_2_	-	1.327 at J_100_	-	-	[84]
	sulfides oxidation	CoFe-LDH	sulfides	0.25 M in MeCN/H_2_O	sulfoxides/H_2_	1.90 at J_5_	1.39 at J_5_	-	-	[87]
		Ni(ii)–bipyridine	phenyl sulfide	H_2_O (30 Equiv.), n-Bu_4_NBF_4_, MeCN	phenyl sulfoxides/H_2_					[89]
	nitroalkanes	NiSe	nitrotoluene	0.4 mM in 1.0 M KOH	E-nitroethene	-	-	1.69	1.36	[12]
	alkane oxidation	Pt	ethylene	1 M KCl	ethylene oxide/H_2_	FE 70%	product specificities 97%			[95]

## 3. Anodic OUR of Organic Compounds in Parallel with CO_2_RR (OUR||CO_2_RR)

The over-combustion of fossil fuels along with excessive anthropogenic emission of CO_2_ acceleratingly exacerbate global environmental and climate issues, which severely hinder the sustainable development of human society. Therefore, efficient conversion of CO_2_ is extremely imperative and urgent to mitigate and circumvent these challenges. With the implementation of renewable electricity, our carbon footprint could be decreased by using CO_2_ electroreduction (CO_2_RR), which can not only serve as a sustainable strategy to curb CO_2_ accumulation, but can also yield high-value-added fuels and chemicals. For a CO_2_ electrolyzer, the OER occurs in the anodic chamber, and large overpotential is required to achieve appreciable current density owing to the sluggish kinetics of the OER. Thermodynamic results suggest that an energy loss of about 90% is attributed to the OER during CO_2_ reduction to CO [29]. Furthermore, O_2_ is less valued, and the ROS caused by the OER may degrade the electrolyzer membrane and thus affect the durability of electrolyzers, as clarified above. Similar to the hybrid water electrolysis, the thermodynamically more favorable OUR integrating with the CO_2_RR (OUR||CO_2_RR) is a promising strategy to yield value-added chemicals on both anode and cathode with lower electricity input into the electrolyzer and exclude the formation of ROS, thus maximizing the utilization of electricity (Table 2) [29,30,31,97,98,99,100,101].

Up to now, the OURs of methanol [99,100], glycerol [29,31,98], 1,2-propanediol [30], hydroxymethylfurfural [98,101], glucose [29] and 2-phenoxy-1-phenylethanol [98] have been tested as anode processes to efficiently integrate with the CO_2_RR. In 2019, Paul J. A. Kenis and co-workers [29] found that the anodic electrooxidation of glycerol can lower the input electricity by up to 53%, which reduced the carbon footprint and operating costs, thereby opening avenues for a carbon-neutral cradle-to-gate process even when the electrolysis is driven by grid electricity (Figure 7a–c). Later in 2020, Erwin Reisner and co-workers [98] reported that glycerol can be selectively electro-oxidized to glyceraldehyde with a turnover number of near 1000 and an FE of 83 %. The cathode yielded a stoichiometric amount of syngas with a CO:H_2_ ratio of 1.25 ± 0.25 and an overall cobalt-based turnover number of 894 with an FE of 82 % (Figure 7d,e). This hybrid electrolyzer of OUR||CO_2_RR inspires the design and implementation of novel strategies for coupling the CO_2_RR to energy-saving, and value-added, oxidative chemistry. Shi group [99] developed a general strategy for an anodic OUR of methanol-to-formic acid integrating with a cathodic CO_2_RR with copper oxide nanosheets on copper foam (CuONS/CF) and mesoporous SnO_2_ on carbon cloth (mSnO_2_/CC) as cathodic and anodic catalysts, respectively. Both sheet-shaped CuONS and mesoporous SnO_2_ can provide significantly large electrochemical surface areas that are key for electrocatalytic reactions. CuONS/CF enabled a low potential of 1.47 V vs. RHE at current density of 100 mA cm^−2^ (Figure 7f), which featured a significantly boosted activity compared to the OER. The mSnO_2_/CC displayed a relatively high FE of 81 % at 0.7 V vs. RHE for formic-acid generation from the CO_2_RR (Figure 7g). A considerably low cell voltage of 0.93 V at 10 mA cm^−2^ was required in the hybrid CuONS/CF||mSnO_2_/CC electrolyzer for formic-acid production at both sides (Figure 7h). As a whole, most of the research on OUR||CO_2_RR were based on model studies, a few were performed in continuous-flow cell, and most of the OURs operated in strongly alkaline solutions. Therefore, there is great room to improve in terms of the electrocatalysts and the optimal operating conditions.

## 4. OUR-Based Fuel Cells or Other Devices

Pioneered by Sir William Robert Grove in 1839, the research and development of fuel cell technology has experienced more than 100 years of growth up to now, and great progress has been achieved. Typically, to maximize the performance of a fuel cell (e.g., output voltage, power density and efficiency), full oxidation of fuel is needed to produce and transfer more electrons. Specifically, for carbonaceous-based fuel cells with excellent electrical performance, usually CO_2_ and or CO_3_^2−^ is generated. So, does this mean that the electrical performance of a fuel cell concomitantly integrated with organic compounds upgrading will be affected? There is no doubt. In fact, the electric performance of the most advanced fuel cells fed by carbonaceous-based fuel (even those including hydrogen) fail to attain the desired end [102,103,104,105]. For example, alcohol-fed fuel cells usually output voltages ranging from 0.5 V to 0.9 V even for H_2_/O_2_ fuel cells, which are at least 0.3 V lower than the theoretical voltage. To upgrade organic compounds and generate electricity profits concomitantly, an optimum balance should and can be kept via fine designed catalysts and systems [11,32]. From the perspective of upgrading organic compounds, the intention of upgrading can be even more inclined.

Up to now, most prevailing electrodes and or photoelectrons focus on the planar structure, which suffers from limited reactants diffusion and sluggish mass transfer, thus resulting in overoxidation of valuable chemicals [106,107]. Recently, microfluidic nanostructures have been reported and fabricated to circumvent the aforementioned drawbacks due to the use of 3-D channels for precise product control and enhanced mass transfer [106,107]. Qu and co-workers [106] constructed a microfluidic photo-electrochemical architecture with 3-D microflow channels, which was fabricated with defect WO_3_/TiO_2_ heterostructures on porous carbon. The charge effectively accumulated on the nanojunction as manifested by the visualizing Kelvin probe force microscopy and photoluminescence spectroscopy (Figure 8a–f). The efficient charge separation contributed to a 3-fold enhancement of the generation of glyceraldehyde and 1, 3-dihydrocxyacetone. Meanwhile, the microfluidic platform with boosted mass transfer exhibited a reaction selectivity of 85%, which was higher than the planar protocol (Figure 8g,h). Moreover, this WO_3_/TiO_2_ heterostructured photoanode enabled effective production of high-value-added KA oil and S_2_O_8_^2−^ via photocatalytic electrooxidation of cyclohexane and HSO_4_^−^ and pollutants degradation (Figure 8i,j). In a photocatalytic fuel cell with the WO_3_/TiO_2_ heterostructured photoanode, a remarkably higher open-circuit voltage of 0.9 V and a short-circuit current of 1.2 mA cm^−2^ were obtained (Figure 8k–o).

Cauê A. Martins and co-workers [107] developed a 3D-printed microfluidic glycerol fuel cell that generates power concomitantly to formate and glycolate. They intelligently tuned the balance between the output energy and the two carbonyl compounds generated via decorating the Pt/C/carbon paper anode in situ and before feeding reactants, or operando with Bi (while feeding reactants). For the operando method, rodlike Bi oxides dendrites were built and were inactive for the glycerol electrooxidation and covered active sites. While the in situ strategy boosted Bi decoration, which can increase the open-circuit voltage to 1.0 V, it augmented the maximum power density 6.5 times and the glycerol conversion up to 72% at ambient conditions. The authors attributed the significant performance to strong CO antipoisoning of the anode, which is conducive to a more complete reaction and harvesting more electrons at the device.

Recently, Huang and co-workers [32] developed ternary Pd–Pt–Ag nanowires for glycerol fuel cells with excellent performance. The ternary Pd–Pt–Ag nanowires were designed via the d band theory, as the-state-of-the-art Pt/C catalysts of a high d band center suffer from severe CO and CO-like intermediates poisoning. Therefore, an optimum d band center was achieved with Pt and/or Pd alloy with Ag with a low d band center. Furthermore, the Pd–Pt–Ag nanowires were also designed based on other significant merits: (i) Pt and Pd exhibit excellent intrinsic activity, (ii) Ag possesses the highest conductivity of 6.30 × 10^7^ Sm^−1^ among all metals at room temperature, (iii) nanowires (NWs) possesses remarkable anisotropy that is particularly in favor of unidirectional electron transfer and anti-aggregation over catalysis compared with nanoparticles and (iv) Ag is used to break C–C bonds and form C1 products. Subsequently, Pd–Pt–Ag nanowires were generated via facile galvanic displacement on the mother Ag nanowires. The as-prepared catalysts displayed high activity and stability of glycerol electrooxidation (2.82 A mg^−1^, 2.03 times higher than that of the commercial benchmark Pt/C catalyst, Figure 9a) and the oxygen reduction reaction (ORR) (Figure 9b). The main product of glycerol electrooxidation on Pd_0.82_Pt_0.56_Ag nanowires was formate with a decent selectivity of 81.2% on Pd_0.82_Pt_0.56_Ag nanowires, and the proposed pathway is glycerol → glyceraldehyde → glycerate → formate (81.2%) and CO_3_^2–^, as manifested by in situ Fourier transform infrared spectroscopy (FT–IR, Figure 9c) and NMR analysis. Surprisingly, a homemade glycerol fuel cell with bifunctional Pd_0.82_Pt_0.56_Ag electrodes delivered a highest-ever-recorded voltage of 1.13 V in glycerol fuel cells (Figure 9d–f), and could achieve a maximum power density of 13.7 mW cm^−2^ under ambient conditions that was ∼50% enhanced compared to that based on the Pt/C benchmark (Figure 9g). The excellent electrocatalytic performance and efficient glycerol upgrading implied a balance between the contradiction of high energy output and the generation of value-added products (Figure 9h,i). Therefore, the OUR of organic compounds in fuel cells provides a novel and fascinating paradigm for the electrosynthesis of high-value-added products, and helps cut our carbon footprint, as conventional carbonaceous-based fuel cells produce greenhouse gas carbon dioxide at the anodes. Notably, we found that the peak-potential of glycerol electrooxidation is near 0.9 V vs. RHE, which is far below 1.23 V, signifying the great potential of Pd_0.82_Pt_0.56_Ag electrodes for the application of glycerol upgrading assisted with hydrogen production. There is no doubt that the key to realizing the organic compounds upgrading is advanced materials, whether in a hybrid electrolyzer for cogeneration of hydrogen or in a fuel cell. Huang and co-workers concluded that decent activity and low oxidation potential of alcohol electrooxidation can be achieved via surface and interface modulation of PGM-based catalysts, such as surface electrochemically etching [44] and surface plasmon resonance [45].

In a conventional fuel cell, the hydrogen of the organic fuels is oxidized to H_2_O under large potential. Subsequently, value-added products and hydrogen are generated at the anode and cathode, respectively. Recently, Wang group [11] found that low-potential furfural electrooxidation drove the electrooxidation of the aldehyde group to generate value-added furoic acid with hydrogen (H) atoms of the aldehyde group to be released as gaseous hydrogen (H_2_) at a low potential of approximately 0 V_RHE_ (vs. RHE). Meanwhile, the integrated electrochemical system can generate electricity of about 2 kWhm^−3^ of H_2_ yield in a cell. This interesting finding may provide a transformative strategy to convert biomass upgrading and H_2_ production from an electricity-input mode to an electricity-output mode.

Besides upgrading organic compounds in fuels, sporadic examples of organic compounds upgrading in other devices were reported, such as in zinc–air batteries. In a conventional zinc–air battery, the discharging process includes zinc electrooxidation on the negative electrode and ORR on the positive counterpart, and the charging process includes the corresponding reversing reactions of Zn^2+^ reduction and oxygen evolution (OER). Unfortunately, the sluggish kinetics of the OER tremendously increase the energy consumption over the charging process. In view of this, recently, Wang and co-workers constructed a zinc–ethanol–air battery with a heterostructured Co(OH)_2_@Ni(OH)_2_ cathode wherein the sluggish OER over the charging process was replaced by the kinetically-favored OUR of ethanol. The resulting charge voltage of the zinc–ethanol–air battery was more than 300 mV lower than the conventional zinc–air battery at the same charging conditions. Furthermore, the elimination of the oxygen bubbles from the OER also guarantees robust charging at high current density.

**Table 2 materials-16-02500-t002:** Electrocatalytic performance of recently reported representative OUR electrocatalysts, the OUR||CO_2_RR systems and OUR-based devices (fuel cell and zinc–air battery).

Working Mode	Reaction Type	Anode Catalyst	Substrate	Electrolyte	Product	3-Electrode System		2-Electrode System		Ref.
						E_OER_ at j_10_ (V_RHE_)	E_OUR_ at j_10_ (V_RHE_)	V_OER-HER_ at j_10_	V_OUR-HER_ at j_10_	
OUR||CO_2_RR on electricity-input mode		CuONS/CF	methanol	1 M KOH	Formic acid	-	1.47 at J_100_	-	0.93	[99]
		CuSn	methanol	1 M in 1 M KOH	formate	1.76 at J_200_	1.49 at J_200_	V_OER-CO2RR_: 3.84 at J_100_	V_MOR-CO2RR_: 3.23 at J_100_	[100]
		PdOx/ZIF-8	HMF	20 mM in 0.5 M [Bmim]BF_4_, 1.0 M CH_3_CN	FDCA	FEco, 97%	FE_organic_ _acid_, 84.3%	-	-	[101]
OUR-based battery on electricity output mode	fuel cells	Pd_0.82_Pt_0.56_Ag	glycerol	1 M in 0.5 M NaOH	formate	-	-	-	V_ocp_: 1.13 V	[32]
		WO_3_/TiO_2_	glycerol	0.5 M H_2_SO_4_	GLA, DHA	-	-	-	V_ocp_: 0.9 V	[106]
		Bi-Pt	glycerol	0.1 M in 0.1 M KOH	glycolate and formate	-	-	-	V_ocp_: 1.0 V	[107]
	Zinc–air battery	Co(OH)_2_@Ni(OH)_2_	ethanol	1.0 M	acetate	-	-	-	-	[33]

## 5. Conclusions and Future Perspectives

In this minireview, we summarized the recent development of the electrochemically oxidative upgrading reaction (OUR) of organic compounds to integrate with the hydrogen evolution reaction (HER) and carbon dioxide reduction reaction (CO_2_RR) in hybrid electrolyzers on electricity-input mode and OUR-based fuel cells and other devices under electricity-output mode to co-produce electricity.

In a hybrid electrolyzer of the HER and or CO_2_RR integrating with the OUR of organic compounds, such as kinetically favored electrooxidation of oxygen-contained molecules (e.g., alcohol, aldehyde, carboxylates and biomass-derived molecules), nitrogen-contained molecules (e.g., amine, nitroalkanes, and tetrahydroisoquinolines), organic sulfides and alkenes were discussed. Compared to conventional water electrolysis that suffers from inherent weaknesses including high theoretical voltages, single value-added products and low atomic efficiency, upgrading the kinetically favored organic molecules to co−generate hydrogen and value-added chemicals leads to products possessing significant theoretical energy efficiency, great atom economy, reduced cell voltage (more energy-saving), no production of ROS and explosive hazards and exogenous-oxidant-free conditions in the anodic chambers. Unlike the sacrificial agent-assisted electrolysis for the hydrogen evolution reaction and or the carbon dioxide reduction reaction, OUR||HER and/or OUR||CO_2_RR via hybrid electrolyzers possess the merits of high theoretical energy efficiency, great atom economy and low price, which seems to be the most fascinating synthetic strategy for hydrogen production, curbing CO_2_ accumulation and organic compounds upgrading, killing two birds with one stone.

In a multifunctional paradigm of batteries with the OUR, value-added chemicals are generated with the co-production of electricity, which can also kill two birds with one stone. Unfortunately, whether for OUR||HER and/or OUR||CO_2_RR, or the OUR of organic compounds in an electricity-output battery, these strategies based on the OUR are still at an early laboratory stage, the corresponding substrates diversities are still limited, the system design still needs to be optimized and the total efficiency is still unsatisfactory.

Therefore, cost-effective catalyst design, reaction mechanism identification, and efficient system designs of electrolyzers and/or batteries (e.g., electrolytes, diaphragms, additives, feeding systems, controlling systems, purification and further application of the products) are highly desirable for future practical developments in the industry. To efficiently upgrade the organic compounds, we provide our perspectives as follows:Future work should focus on theory calculation-guided smart design and the precise synthesis of advanced catalysts. The prerequisite to realize the architecture is the advanced catalysts, especially for the anode oxidation reaction and CO_2_RR. The ideal catalysts must possess high intrinsic catalytic activity, large electrochemical surface areas, maximum utilization of catalytic sites, significant robustness, etc. Therefore, smart design of advanced catalysts is desirable. To screen the best catalysts, machine learning has been demonstrated to be effective. Besides the assistance of theory calculations, advanced synthesis and characterization methods are also crucial to obtain the target catalyst materials. Considering that noble metals are limited to scarce reserves in the Earth’s crust and their resulting high costs, earth-abundant metals and or carbon-based catalysts should be preferentially focused and developed;Future work should work towards the optimization of reaction conditions. Besides the catalyst materials, reaction conditions (e.g., the solvent, additives and temperature) have a great influence on the thermodynamics and kinetics of substrate adsorption and conversion, the target product desorption, on processes such as mass transfer and the microenvironment, and thus on the final catalytic efficiency. Therefore, the optimization of reaction conditions is quite necessary. If conditions render them necessary, theoretical simulations and in situ and/or operando technologies can be helpful;In situ and/or operando technology assisted the characterization and identification on the molecular/electronic level of the active sites, reaction pathways, important intermediates and the final structure-property relationship. In order to explore the advanced catalysts and improve the final catalytic efficiency, the study of the catalytic mechanism is indispensable. In situ and/or operando technologies (e.g., High Performance Liquid Chromatography (HPLC), Differential Electrochemical Mass Spectrometry (DEMS), Fourier-transform infrared spectroscopy (FTIR), Raman Spectra, X-ray powder diffraction (XRD), X-ray photoelectron spectroscopy (XPS) and X-ray absorption fine structure (XAFS)) are powerful tools able to reveal the catalytic mechanism, which conversely guides the optimization of catalysts and reaction conditions;Electrolysis system optimization plays a key role in this research. The state-of-the-art systems also determine efficiency, cost, operability and the safety of these three strategies in both the laboratory and the industry. The design and optimization of systems include but are not limited to the basic cell units (e.g., electrode materials, current collector and diaphragm), feeding units, separation and purification units, controlling system, etc.

With the considerable renaissance of electrosynthesis strategies and the development of renewable-based electricity nowadays, upgrading organic compounds in a multifunctional device would be greatly boosted and more applicable in practical chemical manufacturing if the issues related to the selectivity, activity, mechanism identification, substrate universality, long-term durability of catalysts and the final electrolysis systems were solved.

## Data Availability

Not applicable.

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
