# Peer review of "Killing Two Birds with One Stone: Upgrading Organic Compounds via Electrooxidation in Electricity-Input Mode and Electricity-Output Mode"

_materials, 2023, doi:10.3390/ma16062500_

Round 1

Reviewer 1 Report

1.      How is it possible that a method is either energy-saving or has high electricity utilization? (Page 2, lines 67-68)

2.      On page 2, line 3 it said that ((nitrogen-contained amines)). Is there a mine compound without nitrogen?

3.      Page 3, lines 99-101 need more explanations and a reference.

4.      A concise summary of input/output mode is favorable.

5.      Part 2 (page 3, lines 126-127) needs references.

6.      Page 6, lines 208-211 need references.

7.      Page 12, lines 388-391 need references.

8.      Figure 5 needs some correction. For example, although in part h it is said that the diagram belongs to 1H NMR Spectra, there is the irrelevant diagram, or part i do not exist anymore.

Author Response

Response to referee 1

Comment 1: How is it possible that a method is either energy-saving or has high electricity utilization? (Page 2, lines 67-68)

Response: Thanks for your question. Conventional water electrolysis for hydrogen production suffers from severe sluggish kinetics of anodic oxygen evolution reaction (OER, 1.23 V vs. RHE). Fortunately, organic compounds electrooxidation like alcohols, amines exhibit fast kinetics, for example, the theoretical potential of glycerol electrooxidation is near 0.22 V vs. RHE (doi.org/10.1002/celc.201900502) which is far more below than that of OER. Meanwhile, many articles have demonstrated that organic compounds electrooxidation like alcohols, amines et al. possess a smaller cell voltage than that of the conventional water electrolysis. (J. Am. Chem. Soc. 2021, 143, 29, 10822–10827, Energy Environ. Sci., 2022,15, 5300-5312, ACS Nano 2022, 16, 9572−9582, Nat. Catal. 2022, 5 (3), 185-192.) Therefore, it can be carefully concluded that “the OUR||HER mode requires reduced cell voltages, and is more energy-saving, cost-effective”.

    Furthermore, besides hydrogen generation in the cathode chamber of the OUR||HER electrolyzer, high value-added products can be produced over the process of anodic organic compounds upgrading. While less valued oxygen is produced in the conventional electrolylzer of water. From this perspective, we carefully conclude that OUR||HER possesses high utilization of electricity or electrons over the electrolysis.     

Comment 2: On page 2, line 3 it said that ((nitrogen-contained amines)). Is there a mine compound without nitrogen?

Response: Thanks for your point. We have removed “nitrogen-contained” in front of “nitrogen-contained amines” in the revised manuscript.

Comment 3: Page 3, lines 99-101 need more explanations and a reference.

Response: Thanks for your suggestion. We have added explanations and a reference.

Comment 4: A concise summary of input/output mode is favorable.

Response: Thanks for your suggestion. We have added a summary of input/output mode on page 3, line 118-120 of the revised manuscript.

Comment 5: Part 2 (page 3, lines 126-127) needs references.

Response: Thanks for your point. We have added references on page 4, lines 143 of the revised manuscript.

Comment 6: Page 6, lines 208-211 need references.

Response: Thanks for your suggestion. We have added references on page 7, lines 235 of the revised manuscript.

Comment 7: Page 12, lines 388-391 need references.

Response: Thanks for your suggestion. We have added references on page 12, lines 408 of the revised manuscript.

Comment 8: Figure 5 needs some correction. For example, although in part h it is said that the diagram belongs to 1H NMR Spectra, there is the irrelevant diagram, or part i do not exist anymore.

Response: Thanks for your point and suggestion. We have carefully corrected the caption of Figure 5.

Reviewer 2 Report

 This manuscript reviews the recent progress of electrochemically oxidative upgrading reaction (UOR) of organic compounds which can be combined with HER, CO2RR, and ORR. This is a timely and interesting review, thus I support the acceptance of this manuscript after minor revision.

1) The various UOR examples are well-organized. However, the characteristics of the employed catalysts for each reaction were not sufficiently elucidated. Further explanation is required and this might be helpful for readers to design new catalysts.

Author Response

Response to referee 2

 This manuscript reviews the recent progress of electrochemically oxidative upgrading reaction (OUR) of organic compounds which can be combined with HER, CO2RR, and ORR. This is a timely and interesting review, thus I support the acceptance of this manuscript after minor revision.

Comment 1: The various OUR examples are well-organized. However, the characteristics of the employed catalysts for each reaction were not sufficiently elucidated. Further explanation is required and this might be helpful for readers to design new catalysts.

Response: Thanks for your kind suggestion. We have supplied more explanations about the characteristics of the employed catalysts for each reaction. 

Reviewer 3 Report

The review article summarizes the significant contributions over the last 5-7 years of works demonstrating electrochemical oxidative reactions to transform commodity chemicals, such as alcohols and amines, into value-added chemicals. The article cites recent work from notable publications and summarizes the work sufficiently. However, the manuscript is very difficult to read. It must undergo extensive editing before consideration. I have outlined below a sample of the issues in the first few pages, but errors such as these and many more can be found consistently throughout the manuscript.

In addition to editing, it also needs formatting changes. Specifically, Table 1 is very difficult to understand. It may be better to break it down into multiple tables to make the information more accessible. Also, figure 6g contains references to works that are not present in this review article. Finally, there are numerous acronyms that can become very confusing. The article needs a list of acronyms to help the reader navigate.

Line 12-14: The first sentence in the manuscript does not make sense. It is not a complete sentence.

Line 14: The acronym does not fit grammatically in the sentence.

Line 16: CO2 not CO2

Line 17: enables to?

Line 18-19: I understand that the authors want to use the saying “kill two birds with one stone”, but to put it in a sentence like this with little context is not appropriate for a scientific manuscript.

Line 19-21: I think these sentences should eb combined. Line 19 cannot stand on it’s own.

Line 21: summarize and highlight

Line 22-24: this sentence is poorly written

Line 24: We also

Line 25-26: organic compounds upgrading-based technologies? This description is unclear.

Line 31: makes economy susceptible

Line 32: aggravate greenhouse effect?

Line 33-34: It is strange to see a section in parenthesis followed by -based. I think the parenthesis should be removed and the sentence re-organized.

Line 34: shows

Line 35: incomplete sentence

Line 40: For example

Line 48: enables to store?

Line 52: accepting electron

Line 71: Why is there no need to take price into account for waste removal?

Author Response

Response to referee 3

The review article summarizes the significant contributions over the last 5-7 years of works demonstrating electrochemical oxidative reactions to transform commodity chemicals, such as alcohols and amines, into value-added chemicals. The article cites recent work from notable publications and summarizes the work sufficiently. However, the manuscript is very difficult to read. It must undergo extensive editing before consideration. I have outlined below a sample of the issues in the first few pages, but errors such as these and many more can be found consistently throughout the manuscript.

Response: Thanks for your point and suggestion. We have made extensive editing and correction carefully in the manuscript towards the outlined issues and other similar issues in the manuscript. 

In addition to editing, it also needs formatting changes. Specifically, Table 1 is very difficult to understand. It may be better to break it down into multiple tables to make the information more accessible. Also, figure 6g contains references to works that are not present in this review article. Finally, there are numerous acronyms that can become very confusing. The article needs a list of acronyms to help the reader navigate.

Response: Thanks for your point and suggestion. Table 1 has been broken down into 2 tables in the manuscript. And, Figure 6g and the corresponding caption is removed from the manuscript. Finally, we have added a list of acronyms on page 22, line 679-686.

Comment 1: line 12-14: The first sentence in the manuscript does not make sense. It is not a complete sentence.

Response: Thanks for your point. We have revised the first sentence carefully to “Electrochemically oxidative upgrading reaction (OUR) of organic compounds has gained enor-mous interests over the past few years owing to the advantages of fast reaction kinetics, high conversion efficiency and selectivity et al., which displays great potential to become a key element in coupling with the electricity, synthesis, energy storage and transformation.”

Comment 2: Line 14: The acronym does not fit grammatically in the sentence.

Response: Thanks for your point. We have revised this sentence carefully.

Comment 3: Line 16: CO2 not CO2

Response: Thanks for your point. We have changed CO2 to CO2

Comment 4: Line 17: enables to?

Response: Thanks for your point. We have revised this phrase in the revised manuscript.

Comment 5: Line 18-19: I understand that the authors want to use the saying “kill two birds with one stone”, but to put it in a sentence like this with little context is not appropriate for a scientific manuscript.

Response: Thanks for your suggestion. We have revised this sentence, replaced “two bird can be killed with one stone” with “multiple benefits are to be obtained”

Comment 6: Line 19-21: I think these sentences should eb combined. Line 19 cannot stand on it’s own.

Response: Thanks for your suggestion. We have revised and combined these sentences, the final result is “Although OUR of organic compounds is an old and rich discipline, and is enjoying revival, un-fortunately, this fascinating strategy and its integration with HER or CO2RR, and or electricity generation are still at laboratory stage.”

Comment 7: Line 21: summarize and highlight

Response: Thanks for your point. We have corrected the mistake and used the present tense of verb in the manuscript.

Comment 8: Line 22-24: this sentence is poorly written.

Response: Thanks for your review. We have rewritten this sentence. 

Comment 9: Line 24: We also

Response: Thanks for your point. We have corrected the mistake, and revised in the manuscript.

Comment 10: Line 25-26: organic compounds upgrading-based technologies? This description is unclear.

Response: Thanks for your point. We have corrected the unclear description of “organic compounds upgrading-based technologies” to “OUR coupling with HER or CO2RR in an electrolyzer on electricity-input mode, and or cogeneration of electricity in a primary cell on electricity-output mode”. 

Comment 11: Line 31: makes economy susceptible

Response: Thanks for your point. We have revised “makes” to “make” 

Comment 12: Line 32: aggravate greenhouse effect?

Response: Thanks for your question. Fossil fuels supply about 80% of the word energy, they provide electricity, heat, transportation, and some bulk chemical industry et al.   (https://www.nationalgeographic.com/environment/article/fossil-fuels) Subsequently, large amount of CO2 (greenhouse gas) are produced, namely aggravating the greenhouse effect.    

Comment 13: Line 33-34: It is strange to see a section in parenthesis followed by -based. I think the parenthesis should be removed and the sentence re-organized.

Response: Thanks for your suggestion. We have removed the parenthesis in the revised manuscript.

Comment 14: Line 34: shows

Response: Thanks for your point. We have corrected the corresponding mistake.

Comment 15:Line 35: incomplete sentence

Response: Thanks for your point. We have carefully reorganized this sentence in the revised manuscript.

Comment 16: Line 40: For example

Response: Thanks for your point. We have corrected the mistake.

Comment 17: Line 48: enables to store?

Response: Thanks for your point. We have corrected the mistake in the revised manuscript.

Comment 18: Line 52: accepting electron

Response: Thanks for your point. We have corrected the description in the manuscript.

Comment 19: Line 71: Why is there no need to take price into account for waste removal?

Response: Thanks for your question. Industrial and domestic sewage contains high quantities of harmful pollutants, like, ammonia, urea. The natural decomposition processes are often slow, while electrolysis provides an environmental-friendly strategy to remove the aforementioned wastes (Nano Energy 2022, 104, 107875). Thus, from the perspectives of wastes removal, there is no need to take the price of the sacrificial agent into account. We feel sorry that the corresponding description is not clear, and we have revised this sentence to “there is no need to take the price of the sacrificial agent (waste) into account over the process of environmental remediation.”